# Biometric-Based Key Generation and User Authentication Using Acoustic Characteristics of the Outer Ear and a Network of Correlation Neurons

**DOI:** 10.3390/s22239551

**Published:** 2022-12-06

**Authors:** Alexey Sulavko

**Affiliations:** Department of Comprehensive Information Security, Omsk State Technical University, 644050 Omsk, Russia; sulavich@mail.ru

**Keywords:** biometric template protection, autoencoders, automatic machine learning, correlation analysis, ear canal echograms, protected execution of biometric authentication algorithms

## Abstract

Trustworthy AI applications such as biometric authentication must be implemented in a secure manner so that a malefactor is not able to take advantage of the knowledge and use it to make decisions. The goal of the present work is to increase the reliability of biometric-based key generation, which is used for remote authentication with the protection of biometric templates. Ear canal echograms were used as biometric images. Multilayer convolutional neural networks that belong to the autoencoder type were used to extract features from the echograms. A new class of neurons (correlation neurons) that analyzes correlations between features instead of feature values is proposed. A neuro-extractor model was developed to associate a feature vector with a cryptographic key or user password. An open data set of ear canal echograms to test the performance of the proposed model was used. The following indicators were achieved: EER = 0.0238 (FRR = 0.093, FAR < 0.001), with a key length of 8192 bits. The proposed model is superior to known analogues in terms of key length and probability of erroneous decisions. The ear canal parameters are hidden from direct observation and photography. This fact creates additional difficulties for the synthesis of adversarial examples.

## 1. Introduction

Any unauthorized interference in the work of artificial intelligence (AI) can lead to the following consequences: property damage, information security breaches, threats to life and the health of citizens, technological failures or disasters, etc. All of these depend on the purpose of a particular implementation of AI and its capabilities. Therefore, AI algorithms must support protected execution mode in mission-critical applications. “Protected execution” means the impossibility of analyzing the logic and control of AI and extracting knowledge from the AI memory (for example, personal data) by any unauthorized person.

Responsible AI applications include biometric authentication systems based on fingerprints, the iris, ink, handwriting, voice, and other parameters. Biometric images (patterns) are personal data that need reliable protection against compromise. The “protected execution” of the biometric authentication procedure can be implemented based on homomorphic encryption or by using special mathematical models that make it possible to associate a biometric image (pattern) of a person with his password or personal cryptographic key. The key binding model is usually used to generate a strictly specified key in response to a biometric image of a certain person. The combination of the biometric template and key is stored as helper data. The key binding models can be divided into two main categories: fuzzy extractors (fuzzy commitment, fuzzy vault, fuzzy embedder), which are based on the use of error correction codes, and neuro-extractors (neural fuzzy extractors), which are based on artificial neural networks (ANN).

Each group of models has fundamental disadvantages.

In this paper, we propose a neuro-extractor model based on correlation neurons (c-neuro-extractor). The proposed model has no typical limits compared to existing models. This model has several benefits that allow for the use of longer keys (passwords) for authentication and reduce the false rejection (FRR) and false accept (FAR) rates (the FRR and FAR are expressed as a probability, as in the present work, or as a percentage). Correlation neurons are a new class of neurons that analyze correlations between features instead of analyzing feature values in pattern classification problems. The analysis of the internal correlations of images (patterns) and the classification of the decisions occur without storing the information about the correlations or values of features typical for biometric images of users of computer systems. In other words, the reference information about the class of images is not compromised during its storage. The learning process of correlation neurons is fully automatic and remains robust even on small training sets. The effectiveness of the proposed model is illustrated by the example of the verification of a person by the peculiarities of the ear canal’s internal structure using an open data set of acoustic images (patterns) of ears (AIC-ears-75). The advantage of the ear canal features is that they are not compromised in the natural environment (a photograph of the ear is not informative and is not suitable for making a physical or digital fake of the ear for realizing adversarial attacks) [1]. This problem is difficult to solve because the parameters of the ear canal are significantly less informative than fingerprint and iris parameters.

## 2. Related Works: Common Terms

Homomorphic encryption is a form of perspective encryption that allows one to perform mathematical operations on ciphertext and obtain an encrypted result that matches the result of operations performed in plaintext. The main problem with these methods is poor performance. For example, a comparison of the presented images with fingerprint templates requires significant computational resources. Furthermore, the recognition of a person takes too much time, even when using parallel computing [2]. As mentioned by the authors in [3], the main disadvantage of the proposed homomorphic protection scheme of multibiometric templates is its low performance. The performance degradation is significant, even for simple decision rules. In addition, when recognizing homomorphically encrypted images, the FRR and FAR values often increase. In particular, the authors in [4] proposed a method of homomorphic protection of face parameters extracted from images using deep neural networks. The extracted parameters are encrypted using the Paillier probabilistic cryptosystem. The proposed encryption scheme reduces the accuracy of face verification. The situation can be explained by the fact that homomorphic ciphertexts stop decrypting after performing a sufficiently large number of addition and multiplication operations. Of course, these problems could be solved in the future, but the effective protection of biometric templates is possible without using homomorphic encryption.

The operation of the key binding model requires creating a biometric template and entering a pre-generated cryptographic key or password. This process is performed when a new user registers. A biometric template is a set of reference characteristics of the user’s biometric data (for example, the values of the weight coefficients of a neural network, etc.). The secure template does not compromise the reference biometric characteristics of the user during its storage. Biometric images (patterns) can be represented as raw data, feature vectors, or meta-features. To obtain a feature vector, “raw” biometric data must be processed by a special algorithm. The feature extraction unit converts the raw biometric data into a fixed-length feature vector. The implementation of the feature extraction unit depends on the way in which the biometric images (handwritten signatures, fingerprints, iris images, voice signal spectrograms, EEG, ECG) are presented. A meta-feature is an integral characteristic that is calculated based on the processing of two or more biometric features. The quality of a feature (meta-feature) is determined by its informativeness, that is, the amount of information contained in the feature, which makes it possible to distinguish one user from another.

Key binding models must be resistant to adversarial attacks and knowledge extraction. An adversarial attack consists of subtly modifying someone’s biometric image in order to fool the biometric system. The modified image is called an adversarial image. Knowledge extraction is the receipt by an attacker of the user’s private key or his biometric image, bypassing the security system.

The “Genuine” image is a biometric image (pattern) of a legitimate user. The “Impostor” image is a biometric image (pattern) that does not belong to any of the “Genuine” image classes that are known to the system. Impostor images can be prepared by an attacker for adversarial attacks and other destructive actions. These images are usually attributed to the class “Impostor”.

In fuzzy extractor schemes, the cryptographic key is encoded with error-correcting codes (BCH, Reed–Solomon, Hadamard), then, the key is combined with biometric features and a «secure sketch» is formed. During the authentication process, the user presents biometric data, which are “subtracted” from the secure sketch to obtain the key. If the key contains a small number of bad bits, an error correction algorithm is applied. This approach has a lot of disadvantages. Firstly, the key length in the classical scheme of a fuzzy extractor depends on the correcting ability of the error correction code. The higher the correction capacity, the more redundancy contained in the secure template and the shorter the key length. A fuzzy extractor can work satisfactorily with highly informative biometric parameters, for example, irises (70 bits [5]) or fingerprints (48–64 bits [6]). It is difficult to use the scheme in relation to weak biometric data since in this case, it is necessary to correct a large number of key bits. Secondly, fuzzy extractors are not capable of full-fledged training and do not analyze data like neural networks or other AI algorithms do. When processing weak biometrics, the error probabilities turn out to be very high (for handwritten signatures, the following values of error probabilities were obtained: FAR = 0.0691 with FRR = 0.0785 [7]).

Fuzzy extractors are often combined with pretrained deep neural networks that extract more informative features from the image. The results of such a combination should not be attributed to fundamentally new schemes since they inherit the disadvantages of the fuzzy extractor. The only improvement from such a combination is a more efficient feature extraction unit. In some cases, this may increase the key length. For example, in the work in [8], a key with a length of 128 bits was generated from the parameters of a person’s gait (versus 50 bits in [9], where a fuzzy commitment scheme without a neural network was used in a similar problem).

The first (base) model of a neuro-extractor was developed as the basis for the Russian standard GOST R 52,633.5 [10]. This model was a shallow neural network, consisting of one or two hidden layers, and was trained using automatic learning (without the use of gradient descent). ANNs encode knowledge about the features of biometric images through weight coefficients, which do not provide the direct observation of biometric parameters. The base neuro-extractor model had a threshold neuron activation function to extract the binary code of the key. It was built individually for each user and worked in verification mode (one-to-one comparison), and for training, samples of «Genuine» and «Impostor» images were required, as well as a cryptographic key. The advantage of the base model is that it can be used with any biometric modality. A neuro-extractor allows for the extraction of keys with longer lengths and fewer errors than a regular fuzzy extractor (it was reported that it is possible to extract keys with a length of 256 bits from handwritten signatures [11]). However, the classic neuro-extractor scheme has privacy leak issues. There are some adversarial attacks aimed at extracting knowledge from a neuro-extractor [12]. These attacks are carried out during neuro-extractor execution by observing the statistics of its inputs and outputs. Another attack is called a Marshalko attack [13], which is based on an analysis of the weights and tables of the connections of neurons.

Finally, models of deep neuro-extractors based on multilayer convolutional neural networks (CNNs) have been proposed for applications of facial biometrics. The ANN in [14] included two convolutional layers, a max-pooling layer, two fully connected layers, and two dropout layers. The following values for the error rates were obtained: FAR = 0.01 with FRR = 0.0241 (with a key length of 1024 bits). In [15], an ensemble (stack) of two neural networks was used. The first VGG-Face network (13 convolutional and 2 fully connected layers) was pre-trained on 2.6 million images and received 224 * 224 face images as inputs, and produced a 4096-bit binary code at the output. The second network (six fully connected layers trained by the Adam optimizer) translated a 4096-bit vector into a user key, which was specified during training and had a length of up to 1024 bits. The equal error rate was EER = 0.036 (EER = FAR = FRR). The disadvantage of deep fuzzy neuro-extractor schemes is that gradient descent algorithms tend to overfit. The scheme can be poorly transferable to other modalities since the image structure and ANN architecture are different for each modality. The proposed approach is difficult to apply in cases where collecting a large amount of training data is impossible. The benefits of applying such security schemes to facial biometric images are not entirely clear since face parameters can be easily compromised by photography.

Classifiers based on CNNs with the softmax function at the output are vulnerable to adversarial attacks [16]. It is known that the imposition of additive Gaussian noise on the image significantly increases the FAR when verifying the biometric images using neural networks with a similar architecture [17]. If an attacker has access to the weight coefficients, it greatly simplifies the process of carrying out these attacks. Therefore, the neuro-extractor architecture should be built so that the synapse weights do not compromise the biometric data of users.

## 3. Materials and Methods

In the proposed scheme, a feature extraction unit and a c-neuro-extractor are highlighted (Figure 1). The requirement for the features is that each feature extracted from an image must obey the normal distribution law or, at least, the feature probability density function must be symmetric and unimodal.

In the present study, an autoencoder based on convolutional neurons was used. The autoencoder is an artificial neural network that can compress the dimension of the input data, encoding them with a set of informative features, as well as recovering the input data from the feature vector. The autoencoder consists of two subnets. The encoder is used only to extract features. A decoder is used for data recovery (it is impossible to train the encoder without the decoder).

The autoencoder can be trained on a large dataset of anonymized biometric images. The encoder can remain in an unprotected form after training since it does not produce classification solutions at its outputs and does not store personal biometric data and user keys. The encoder outputs (features) must be converted to Bayes–Minkowski meta-features using a special mapping (Figure 1). Then meta-features must be connected to c-neuro-extractors. A separate c-neuro-extractor, which is trained on «Genuine» and «Impostors» images in a trusted environment, is created for each new user. It takes the place of the softmax layer and is capable of generating an almost random output in response to an ambiguous «Impostor» image or user key in response to a «Genuine» image.

C-neuro-extractors can be placed anywhere after being trained. The encoder can be placed in the cloud so feature extraction functions are available to all users. In this approach, the decoder can be removed.

The proposed architecture of the authentication system combines the advantages of deep neural networks (the ability to extract highly informative features and use transfer learning) with the advantages of c-neuro-extractors (protection of private keys and biometric personal data from compromise, robust automatic learning on a small sample set of user images). The c-neuro-extractor model based on correlation neurons and Bayes–Minkowski meta-features space is proposed.

### 3.1. Theory

#### 3.1.1. A Curved Feature Space: The Informativeness and Cross-Correlation of Features

The Euclidian proximity measure and Manhattan distance [18,19] were used as the decision rules, which were protected by homomorphic encryption. These proximity measures were generalized in the form of the Minkowski measure [20] (1):(1)$$y=\sqrt[{p}]{\sum_{j=1}^{n}{\left|\frac{\left(m_{j}-a_{j}\right)}{\sigma_{j}}\right|}^{p}},\bar{a}=\{a_{1,},\ldots ,a_{n}\},p>0$$
where *ā—*is a feature vector representing a biometric image; *a_j_* is the value of the *j-th* feature from the vector *ā*; *n* is the number of features; *m_j_* and *σ_j_* are the mathematical expectation and standard deviation of the values of the *j-th* feature for the «Genuine» class, which is compared with the image *ā* (the «Genuine» class represents the biometric images of one of the legitimate users); *p* is a power coefficient that determines the level of “curvature” of the space. For *p* = 1, we obtain the Manhattan measure; for *p* = 2, the Euclidian measure; and for *p*→∞, the Minkowski measure tends to the Chebyshev measure. The Minkowski distance changes depending on the value of the power coefficient *p*. Figure 2 illustrates what a circle might look like in a two-dimensional Minkowski space. In contrast to the example in Figure 2, the feature space is multidimensional (each feature “creates” one dimension).

The curvature of the feature space occurs due to the correlations between its dimensions (Figure 3). Typically, the feature space is neither flat nor equally curved. All classes of images have different matrices of correlation coefficients *C_j,t_* (2) between features (the biometric image of each person has a unique correlation matrix). Therefore, the feature space is curved differently for the various classes of biometric images.
(2)$$\begin{array}{l} C_{j,t}=\frac{\sum_{k=1}^{K_{G}}\left(a_{t,k}-m_{t})(a_{j,k}-m_{j}\right)}{\sqrt{{\sum_{k=1}^{K_{G}}\left(a_{t,k}-m_{t}\right)}^{2}\sum_{k=1}^{K_{G}}{\left(a_{j,k}-m_{j}\right)}^{2}}}\approx \\ \frac{K_{G}\sum_{k=1}^{K_{G}}(a_{t,k}-a_{j,k})-(\sum_{k=1}^{K_{G}}a_{t,k}\cdot \sum_{k=1}^{K_{G}}a_{j,k})}{\sqrt{\left(K_{G}\sum_{k=1}^{K_{G}}a^{2}{}_{t,k}-(\sum_{k=1}^{K_{G}}a_{t,k}{)}^{2}\right)\left(K_{G}\sum_{k=1}^{K_{G}}a^{2}{}_{j,k}-(\sum_{k=1}^{K_{G}}a_{j,k}{)}^{2}\right)}}, \end{array}$$
where *K_G_* is the number of images in the «Genuine» training set (*K_I_* is the number of images in the «Impostors» training set), and *k* is an index of the images in the «Genuine» training set.

The informativeness level of a feature is an important indicator [21]. The amount of individual information of the *j*-th feature for a certain class of images is determined using Formula (3):*I_j_* = −log_2_(*AUC*(Փ_G_(*a_j_*), Փ_I_(*a_j_*))),(3)
where *AUC, the* area under the curve, is limited by the probability density functions «Genuine» *Փ_G_(a_j_)* and «Impostors» *Փ_I_(a_j_)*, as well as by the *x*-axis. Փ_G_(*a_j_*) characterizes the values of the feature strictly for a certain class of images, and *Փ_I_(a_j_)* characterizes the values of the same feature for all classes of images as a whole [21]. The higher the *I* on average, the further separated the proper class regions in the feature space.

The number of classification errors when using the Minkowski measure can be decreased by changing the parameter *p* that was demonstrated in [20]. The optimum value of *p* depends on the average indicators of the information content and intraclass correlation between features (*I* and *C*, respectively).

However, the correlation not only warps the feature space but also transfers some of the information about the images to “hidden” dimensions. This information can be used as features for image classification, which is shown for the first time in the next paragraph.

#### 3.1.2. The Bayes–Minkowski Meta-Feature Space

To extract information about the levels of curvature of the feature space in the direction of each dimension, we introduce several variations of the Bayes–Minkowski measure (4)–(9), which operate using the differences between the features. These metrics take the smaller values and the higher *C_j_,_t_* (Figure 4). If the *t-th* and *j-th* features are linearly dependent (*C_j_,_t_ = 1*) for images of a certain class, then the *t-th* and *j-th* dimensions with respect to this class become “singular” (combined into one dimension, Figure 3c) and the corresponding difference under the modulus sign always takes the value of zero (as if there is no *j-th* dimension). However, if the features have a weak dependence, the difference (in modulus) increases. The higher the correlation between the features, the lower the percentage of wrong decisions that will be obtained. Figure 4 shows that *AUC_|C|_*
_> 0.__95_(Փ_G_(*y*), Փ_I_(*y*)) < AUC*_|C|_*_< 0.__3_(Փ_G_(*y*), Փ_I_(*y*)).
(4)$$y_{t}=\sum_{j=1}^{n}\left|{\left|\frac{\left(m_{t}-a_{t}\right)}{\sigma_{t}}\right|}^{p}-{\left|\frac{\left(m_{j}-a_{j}\right)}{\sigma_{j}}\right|}^{p}\right|,p>0,j\neq t,$$
(5)$$y_{t}=\sqrt[{p}]{\sum_{j=1}^{n}\left|{\left|\frac{\left(m_{t}-a_{t}\right)}{\sigma_{t}}\right|}^{p}-{\left|\frac{\left(m_{j}-a_{j}\right)}{\sigma_{j}}\right|}^{p}\right|},p>0,j\neq t,$$
(6)$$y_{t}=\sum_{j=1}^{n}\left|{\left|\frac{\mu_{t}-a_{t}}{\delta_{t}}\right|}^{p}-{\left|\frac{\mu_{j}-a_{j}}{\delta_{j}}\right|}^{p}\right|,p>0,j\neq t,$$
(7)$$y_{t}=\sqrt[{p}]{\sum_{j=1}^{n}\left|{\left|\frac{\mu_{t}-a_{t}}{\delta_{t}}\right|}^{p}-{\left|\frac{\mu_{j}-a_{j}}{\delta_{j}}\right|}^{p}\right|},p>0,j\neq t,$$
(8)$$y_{t}=\sum_{j=1}^{n}\left|{\left|\frac{a_{t}}{\delta_{t}}\right|}^{p}-{\left|\frac{a_{j}}{\delta_{j}}\right|}^{p}\right|,p>0,j\neq t,$$
(9)$$y_{t}=\sqrt[{p}]{\sum_{j=1}^{n}\left|{\left|\frac{a_{t}}{\delta_{t}}\right|}^{p}-{\left|\frac{a_{j}}{\delta_{j}}\right|}^{p}\right|},p>0,j\neq t,$$
where *μ_j_* and *δ_j_* are normalizing coefficients calculated as the mathematical expectation and standard deviation of the feature values for the «Impostors» class. The meaning of the *μ_j_* and *δ_j_* coefficients is to bring all the features to an approximately single scale; *μ_j_* and *δ_j_* do not compromise the data of any user since they represent the parameters of the distribution of the feature values for a set of depersonalized images. Thus, differential confidentiality is ensured using Measures (6)–(9). Measures (8)–(9) provide the highest level of confidentiality since they operate only using the normalizing coefficients of the range of *δ_j_*; therefore, their use is preferable for solving the considered problems. To enhance the confidentiality of Measures (6)–(7), noise can be added to *μ_j_* (random shift of the value).

The meta-features are the differences and have the following forms (10)–(12):(10)$${a^{\prime }}_{j^{*}}={a^{\prime }}_{t,j}=f(a_{t},a_{j})=\left|{\left|\frac{\left(m_{t}-a_{t}\right)}{\sigma_{t}}\right|}^{p}-{\left|\frac{\left(m_{j}-a_{j}\right)}{\sigma_{j}}\right|}^{p}\right|,j>t,j^{*}=\sum_{t^{*}=1}^{t-1}(n-t^{*})+j-t$$
(11)$${a^{\prime }}_{j^{*}}={a^{\prime }}_{t,j}=f(a_{t},a_{j})=\left|{\left|\frac{\mu_{t}-a_{t}}{\delta_{t}}\right|}^{p}-{\left|\frac{\mu_{j}-a_{j}}{\delta_{j}}\right|}^{p}\right|,j>t,j^{*}=\sum_{t^{*}=1}^{t-1}(n-t^{*})+j-t$$
(12)$${a^{\prime }}_{j^{*}}={a^{\prime }}_{t,j}=f(a_{t},a_{j})=\left|{\left|\frac{a_{t}}{\delta_{t}}\right|}^{p}-{\left|\frac{a_{j}}{\delta_{j}}\right|}^{p}\right|,j>t,j^{*}=\sum_{t^{*}=1}^{t-1}(n-t^{*})+j-t$$

These differences are a rough (point) estimate of the correlation dependence between the two initial features under the numbers *j* and *t* (the smaller *a’* in the modulus means a higher intraclass correlation between relevant features if *C_j,t_* = 1, then *a’_t,j_* ≈ 0). A point estimate is an estimate made from just one sample of biometrics but in the presence of some a priori knowledge (*m_j_, σ_j_, δ_j_, μ_j_*) obtained in the training process.

The dimension of the Bayes–Minkowski meta-feature space is (13):*n’* = 0.5(*n*(*n* − 1)) = 0.5*n^2^* − 0.5*n*, *n* > 0(13)

Measures (4)–(9) are linear classifiers in the Bayes–Minkowski meta-feature space. We can transform the initial feature space into the rectifying Bayes–Minkowski meta-feature space using mapping *a’_t,j_ = f(a_t_, a_j_)* to use any classifiers. It is most convenient to depict three-dimensional initial and rectifying spaces (Figure 5) since *n ‘ = n = 3*. The space of the meta-features can contain much more information about the classes of images than the initial space. Figure 5 shows that in the initial space, the classes are linearly inseparable and the informativeness of the features is very low (0.1 < *I_j_* < 0.2) but the meta-features are much more informative (0.35 ≤ *I’_j^∗^_* ≤ 2.95). The initial features are highly correlated (0.94 ≤ *C_j,t_* ≤ 0.96), whereas the correlation between the meta-features is insignificant (0.1 ≤ *C’_j^∗^,t^∗^_* ≤ 0.22).

Negatively correlated initial features can form correlated pairs of meta-features. Figure 6 illustrates two positively correlated features (normalized to *δ_j_*) forming a meta-trait *a’_2_* with chaotic dynamics (which has no significant correlation with other meta-features). Negatively correlated features form meta-features *a’_1_* and *a’_3_*, which are positively correlated (relative to each other) for class 1 and negatively correlated for class 3. For class 2, *a’_1_* and *a’_3_* have an implicit correlation—at a certain moment, the positive correlation changes to a negative one.

In order to eliminate the negative correlation, one of the mappings *a’_t,j_ = f(a_t_, a_j_)* should be used repeatedly (first, in relation to pairs of negatively correlated features, then, pairs of positively correlated meta-features). The meta-features space of the second order (after the repeated “transition”) even has a greater dimension (14):*n*’’ = 0.5(*n’*(*n’* − 1)) = 0.5*n*’^2^ − 0.5*n*’ = 0.5(0.5*n*^2^ − 0.5*n*)^2^ − 0.5(0.5*n*^2^ − 0.5*n*), *n* > 0, *n’* > 0(14)

For this reason, the informativeness of the negative correlations of the initial features can be significantly higher than the positive ones. Further “transition” (construction of meta-feature spaces of the third, fourth, and other orders) makes sense until there are correlated pairs of meta-features. The correlation between all pairs of meta-features becomes weak on the Chaddock scale usually after two-three “transitions”

Thus, when constructing classifiers, meta-features obtained from pairs of positively and negatively correlated features can be used. At the same time, the use of meta-features generated by pairs of independent (weakly correlated) features (or meta-features) should be avoided since such generations can be noisy. Independent (weakly correlated) features should be processed separately, without performing transformations over them Equations (10)–(12).

The “naive” scheme of the Bayes classification is fully correct when features are independent, i.e., the feature space has absolutely no curvature. Minkowski’s measure, on the contrary, measures the distance in the curved space. New metrics convert the space of correlated features into the space of independent meta-features so they are named using Bayes–Minkowski metrics.

#### 3.1.3. Assessment of Bayes–Minkowski’s Meta-Feature Informativeness Using Synthetic Datasets

A computational experiment was conducted on the recognition of images (patterns) in the space of abstract (imitated) features. All features had a normal distribution of values. A total of 65 classes of images in spaces of independent and dependent features with different *I* indicators were generated. The generated classes of the images differed by the parameters of the feature distribution. The method of feature and image (patterns) generation was based on the Monte Carlo method and is described in [20].

To identify the generated images on a closed set of 65 classes, a computational experiment was provided with the use of the “naive” Bayesian classifier. For the training of the Bayesian classifier, the parameters of the normal law distribution were calculated (mathematical expectation and standard deviation) for each feature or meta-feature according to the training set (10 random samples of images per class). Conventional probability densities were calculated in accordance with the normal distribution law. The 100 images from each class that were not included in the training set were used as the testing set. The decision was taken in favor of the hypothesis with the highest a posteriori probability. Rank 1 accuracy was calculated (the number of correct classification decisions was divided by the total number of experiments). The test results are presented in Figure 7. This experiment demonstrates the following:A correlation between features can carry more information than the features themselves. If the initial features are more informative (*I* ≈ 0.5) and independent (|*C_j,t_*| < 0.3), then in the meta-feature space, the accuracy of the identification of images is lower than in a case where the initial features are less informative (*I* ≈0.15) but strongly correlated (*1 > C_j,t_ > 0.95*).If the initial features are independent, the meta-features cause noise (the accuracy of the identification is higher when using only the initial independent features than when combining the independent features with the meta-features);The “transition” to the space of the meta-features does not lead to a manifestation of the “Curse of dimensionality” problem if the features are strongly correlated. The curse of dimensionality is a problem associated with an exponential increase in the volume of the training set and related calculations due to the linear growth of the dimension of the features. As we can see in Figure 7, by using a similar training set (10 images), it is possible to achieve higher accuracy when going into the space of greater dimension (*n’* = 435) compared to the initial dimension (*n* = 30). At the same time, the number of calculations when computing a posteriori (“Bayesian”) probabilities grows linearly to the increase in the dimension of the feature space. The number of calculations when computing the correlation matrix and the number of features grows not exponentially but according to a power law (10).

From the simulation results, it can be seen that the optimum according to the accuracy of the recognition of images was achieved at 0.7 ≤ *p* ≤ 1 (depending on the level of the features’ informativeness). When using the mapping in (11), a significant increase in accuracy was not observed.

An assessment of the meta-features’ informativeness in relation to the informativeness and pair correlation of the initial features (Figure 8) was also carried out. The extremum of the average informativeness of the meta-features was observed at 0.9 ≤ *p* ≤ 1. The setting of the value *p* to 0.9 is preferable because the transformation of the features to the meta-features is nonlinear at *p* ≠ 1. If initial features are independent, then *I’* < *I*. However, if features are correlated, then *I’* > *I*. The correlation relationship between two features may be more informative than the pair of features itself. For example, when 1 > *C* > 0.95 and *I* = 0.15, then *I* = 0.488 (at *p* = 0.9).

The described regularities are valid when features have a normal distribution. For other distribution laws, evaluations were not provided.

So, the higher the level of correlation, the more informative the Bayes–Minkowski meta-features. The obtained results show that the Bayes–Minkowski meta-feature space is the best approach to use, at least in the task of pattern classification.

### 3.2. C-Neuro-Extractor

#### 3.2.1. Correlation Neuron Model for Biometric Authentication

Each neuron should separate the input data according to the level of correlation. The neuron is connected to the meta-features that were generated by features paired with a similar level of mutual correlation. Let us introduce two levels of correlations of features: *C_-_* > *C_j,t_* (*C_-_*ϵ[−0.99;−0.3]) and *C_+_* < *C_j,t_* (*C_+_*ϵ[0.3; 0.99]). With |*C_-+_*| < 0.3, the correlation neuron may work incorrectly and the number of errors will be significant. Condition |*C_-_*| = |*C_+_*| does not have to be fulfilled; the more negatively and positively correlated pairs of features, the higher the absolute values of the threshold coefficients *C_-_* and *C_+_* should be set. One meta-feature should be associated with only one correlation neuron in order to avoid the implementation of Marshalko attacks [13]. Thus, correlation neurons are partially connected.

The metric in (15) works well with positively correlated data but cannot determine negatively correlated data (Figure 4). The metric of the standard deviation of the meta-feature values in (16) allows for the separation of both positively and negatively correlated data (Figure 9), which is conditional on the fact that the values of the deviation modules |*a’_j^∗^_* − *m’*| tend to reduce if the correlation between the initial features is strong (both positively and negatively).
(15)$$y=\sum_{j^{*}=1}^{n^{\prime }}{a^{\prime }}_{j^{*}},$$
(16)$$y=\sqrt{\frac{1}{n^{\prime }}\sum_{j^{*}=1}^{n^{\prime }}{\left({a^{\prime }}_{j^{*}}-m^{\prime }\right)}^{2}},m^{\prime }=\frac{1}{n^{\prime }}\sum_{t^{*}=1}^{n^{\prime }}{a^{\prime }}_{t^{*}},$$

Correlation neurons can be based on the weighted standard deviation metric in (17):(17)$$y=\sqrt{\frac{1}{\eta }\sum_{j^{*}=1}^{n^{\prime }}w_{j^{*}}{\left({a^{\prime }}_{j^{*}}-m^{\prime }\right)}^{2}}=\sqrt{\frac{1}{\eta }\sum_{\iota =1}^{\eta }w_{\iota }{\left({a^{\prime }}_{\iota }-m^{\prime }\right)}^{2}},m^{\prime }=\frac{1}{\eta }\sum_{\iota =1}^{\eta }{a^{\prime }}_{\iota },$$
where *y* is a neuron response, *η* is the number of neuron inputs, *w_j^∗^_* is the weight of the synapse under the number *j^∗^* (*w_j^∗^_* ≥ 0, if *w_j^∗^_* = 0, then the *j^∗^-*th meta-feature does not affect the sum, i.e., it does not connect to a neuron), and *ι* is the number of meta-features without taking into account synapses with zero weight. Metric (17) realized a “transition” into the space of the Bayes–Minkowski meta-features of the second order (*a’’_ι_* = (*a’_ι_*–*m’*)^2^) but only for the neuron-related meta-features. The synapse weight is calculated using Formula (18):(18)$$w_{\iota }=\frac{\left|m_{(G),\iota }^{\prime \prime }-m_{(I),\iota }^{\prime \prime }\right|}{\sigma_{(G),\iota }^{\prime \prime }\cdot \sigma_{(I),\iota }^{\prime \prime }},$$
where *m_(G),ι_’’*, *m_(I),ι_’’* are the mathematical expectations, and *σ_(G),ι_’’*, *σ_(I),ι_’’* are the standard deviations of the values of the *ι*-th meta-feature of the second order (*a’’_ι_* = (*a’_ι_* − *m’*)^2^) for the «Genuine» and «Impostors» images. Parameters *m_(G),ι_’’*, *m_(I),ι_’’*, *σ_(G),ι_’’*, *σ_(I),ι_’’* must be deleted after training.

It is proposed to use the multilevel threshold quantization function in (19) as an activation function:(19)$$\phi (y)=\{\begin{array}{c} 3,y<T_{left}\\ 2,T_{left}\leq y<T_{middle}\\ 1,T_{middle}\leq y<T_{\mathrm{right}}\\ 0,y\geq T_{\mathrm{right}} \end{array}$$
where *ɸ(y)* is a neuron output and *T_left_*, *T_middle,_* and *T_right_* are the left, middle, and right threshold values of the neuron activation (Figure 9). In accordance with the proposed model, a neuron has four activation outcomes {0, 1, 2, 3} and only one of them corresponds to the «Genuine» hypothesis; the rest correspond to the «Impostors» hypothesis. The potential attacker does not have information about the correct activation state that corresponds to the «Genuine» hypothesis (herein, *ɸ_G_*) since it is not saved after the neuron is configured.

The purpose of neuron training is that a certain state almost always appears at the output of the neuron when the «Genuine» images enter the neuron; in other cases, the states {0, 1, 2, 3} at the output of the neuron become equally probable: P (0) ≈ P (1) ≈ P (2) ≈ P (3) ≈ 0.25. The P(*ɸ(y)*) is the relative frequency of the occurrence of *ɸ*(*y*) when the «Impostor» image enters at the input of the neuron. It is difficult to achieve such an exact ratio in practice. To provide a high entropy of neuron outputs in response to the «Impostors» images, it is sufficient to adhere to the following ratio: 0.1 < P(*ɸ(y)*) < 0.4.

When the thresholds were being calculated, the probable boundary values of the responses of the neuron *y* to the «Genuine» (*y_Gmin_*, *y_Gmax_*) and «Impostors» (*y_Imin_*, *y_Imax_*) training samples were first calculated using Formula (20). Then, the values of the corresponding distribution functions *F_G_(y)* and *F_I_(y)* were calculated using Formula (21), and the probability density was calculated using Formula (22). For the first approximation, the distribution law of the random variable *y* in (17) was close to the normal one in (20), which was confirmed by the Chi-square method on a large set of the generated data.
*y_min_* = *ξ* − *4 ς*, * y_max_* = *ξ + 4 ς*,(20)

(21)$$F(y)=\int_{-\infty }^{y}\Phi (\zeta )d\zeta ,$$(22)$$\Phi (y)=\frac{1}{\varsigma \sqrt{2\pi }}e^{-\frac{{\left(y-\xi \right)}^{2}}{2\varsigma^{2}}},$$
where *ξ* and *ς* are, respectively, the mathematical expectation and the standard deviation of the *y* values that were calculated based on the training set.

It was proposed to set the thresholds in accordance with the algorithm, which is illustrated in Figure 10. We also introduced the AUC_MAX_ coefficient equal to the maximum allowable AUC(Փ_G_(*y*), Փ_I_(*y*)) for a neuron in order to exclude “weak” neurons that give close responses to the «Genuine» and «Impostors» images (Figure 10).

One of the hash transformations should be applied to modify the value of the activation function (Table 1). A hash transformation should be chosen randomly during neuron training but considering which two key bits (hereinafter *b*) the neuron should be set to. For example, if *ɸ_G_* = 1 and *b* = “10”, the hash transformation number was selected from the set {5, 6, 9, 10, 15, 16, 21, 22} (Table 1).

So, it is sufficient to determine the associated meta-features, calculate weights and thresholds, and set a hash transformation to train a correlation neuron (Figure 10 and Appendix A).

#### 3.2.2. Synthesis and Automatic Training of C-Neuro-Extractors

A c-neuro-extractor is a shallow neural network consisting of one hidden layer of correlation neurons. It works with feature vectors so “raw” images of the «Impostors» and «Genuine» training sets must first be processed with the encoder (Figure 1).

The mapping in (12) at *p* = 0.9 was used to transform the features into meta-features. The normalization coefficients *δ_j_* for switching to the space of the meta-features must be calculated based on the «Impostors» training set before building and learning the c-neuro-extractors.

A case is further considered when the number of inputs *η* for all correlation neurons should be equal. With the synthesis of the c-neuro-extractor, it is necessary to make sure that there is a sufficient number of pairs of features and levels of mutual correlation *C_j,t_* < *C_-_* and *C_j,t_* > *C_+_*. To do this, it is necessary to calculate the correlation matrix according to the «Genuine» training set. Any pair of correlated features potentially generates one meta-feature. Let *N*_-_ and *N*_+_ be the number of neurons focused on processing the levels of correlated data of *C_j,t_* < *C_-_* and *C_j,t_* > *C_+_*, respectively. The condition *N*_-_ ≈ *N*_+_ should be observed (discrepancy is allowed by 1–3 neurons). Each neuron must handle a unique combination of meta-features and generate 2 bits at the output. The required number of neurons is determined based on the required key length *L*. For practical purposes, a sufficient length is *L* = 1024 bits, then *N*_-_ = *N*_+_ = *L*/2/2 = 256. Then, if *η* = 4, 2048 pairs of features will be required (1024 = 256 · 4 pairs for each level of correlation) for the synthesis of the c-neuro-extractor. When using autoencoders, the number of features can be made arbitrary. For example, 256 features give 32,640 potential pairs, of 2048 (≈6.27%) are chosen.

The proposed algorithm of the synthesis and training of the c-neuro-extractor can be summarized as a sequence of steps:Calculation of the feature correlation matrix.Counting pairs of negatively correlated features (*C_j,t_ < C_-_*). If the number of pairs is less than *η·N_-_*, then *C_-_* is increased by 0.05 and this step is repeated.Synthesis and training of *N_-_* neurons for the analysis of negatively correlated data in accordance with the algorithm in Figure 10. If the number of neurons satisfying the conditions of the algorithm in Figure 10 turns out to be less than *N_-_*, then *C_-_* increases by 0.05 and steps 2–3 are repeated.Counting pairs of positively correlated features (*C_j,t_ > C_+_*). If the number of pairs is less than *η·N_+_*, then *C_+_* decreases by 0.05 and this step is repeated.Synthesis and training of *N_+_* neurons for the analysis of positively correlated data in accordance with the algorithm in Figure 10. If the number of neurons satisfying the conditions of the algorithm in Figure 10 is less than *N_+_*, then *C_+_* decreases by 0.05 and steps 4–5 are repeated.

As the interval (*C_-_*; *C_+_*) narrows, it is permissible not to delete already created neurons (new neurons can be added to existing ones). The algorithm is executed until the condition *N_-_ = N_+_ = L/4* is fulfilled or until the condition *C_-_* ≤ 0.3 ∨ *C _+_* ≥ 0.3 is violated. The latter means that it is not possible for the user to associate the c-neuro-extractor with the key length *L*.

Weight coefficient tables and the numbers of hash transformations of the trained correlation neurons represent the secure template of the user.

A c-neuro-extractor has the following peculiarities:correlation neurons are not affected by the problem of imbalance in training (the size of the «Impostors» training set is much larger than the size of the “Genuine” training set);the correlation network setting up process is robust (overfitting does not occur);the length of the key associated with the c-neuro-extractor is potentially much larger than those associated with the fuzzy extractors and the base model of the neuro-extractor;this model should have a much higher level of resistance to adversarial attacks [16,17] than the classical deep network with the softmax activation function at the output, at least in terms of the effect on the FAR indicator. Adding noise and other modifications is unlikely to affect the closeness of the correlations of the «Impostor» image to the «Genuine» image.

In this work, two intervals of correlation of features (*1 > C_j,t_ > C_+_ and −1 < C_j,t_ < C_-_*) and 4 intervals of quantization for the activation function are considered (19). Increasing the number of intervals should enhance the hashing properties of the c-neuro-extractor.

## 4. Experiments and Results

### 4.1. Data Set

A biometric authentication method based on an analysis of the acoustic properties of the ear canal was proposed in [1] and uses headphones with a built-in microphone. The headphones produce a sound signal that resonates in the ear canal. Since the inner ear structure for each person has individual characteristics, the parameters of the propagating sound signal in each ear change in different ways. The microphone records the reproduced signal whose parameters can be considered biometric features. The structure of the ear canal does not change significantly after eight years of age [22].

Traditional biometric parameters, such as fingerprints, face images, signatures, voices, etc., are compromised in the natural environment. These parameters can be “intercepted”, for example, fingerprints can be removed from door handles, mugs, and photographs and voices can be recorded on a dictaphone. The advantages of the method in [1] are that the characteristics of the auditory canal are hidden from direct observation and cannot be copied by being photographed. A “flat” ear image is not informative enough to make adversarial copies.

In the previously mentioned study [1], a set of impersonal (depersonalized) data of ear canal echograms of 75 computer users aged 18–40 years (AIC-ears-75) was collected, which is available for research purposes. Each echogram (or acoustic image of the auditory canal) is presented as a wav format file (mono, 44 kHz, 16 bit). Fifteen measurements of each ear were made for each user. After each measurement, the user removed and put on the headphones again (a device in the form of headphones with a built-in microphone for recording the reflected signal). Each user had to listen to a mono sound signal of increasing and decreasing frequencies (sliding modulated sine), obtained using linear frequency modulations (chirp signals). The signal frequencies varied in the range of 1 kHz to 14 kHz and the signal duration was 10 s (5 s frequency increases, 5 s decreases). The dataset includes 2 folders for the right and left ears with each folder containing 75 subfolders of the ear measurements of the related users.

### 4.2. Image Preprocessing

For the preliminary processing of the acoustic images of the auditory canal, the method of calculating the so-called averaged signal spectrum proposed in [1] was used. To obtain the averaged amplitude spectrum using a short-time Fourier transform, we first calculated the spectrogram with a window size *W_size_* = 65,536 (about one-and-a-half seconds of the signal according to [1]) and a step size *W_step_* = 16,384 (2 times less than that used in [1]). Further, for all windows, the spectrum of the average values of the amplitudes was calculated depending on the frequency (Figure 11a). The first 1500 and last 3000 samples were removed from the averaged spectrum in order not to take into account frequencies less than 1 kHz (this is the initial frequency of the chirp signal) and more than 20 kHz (the microphone was not able to register these frequencies). Frequencies of 14–20 kHz were considered since useful information can appear in overtones. Then, the obtained averaged spectra were “compressed” to 2048 samples using the linear interpolation algorithm. Before being fed into the neural network, the images were reduced to the range of values [0; 1].

In this work, the following window functions were used to obtain averaged spectra: Hamming, Blackman, triangular (Bartlett), rectangular, Gaussian (standard and parametric (generalized) with a shape parameter value of 1.5), and Laplace. Various features were extracted depending on the applied window function.

### 4.3. Autoencoder Training for Feature Extraction

To build feature extraction units, it was decided to use two similar autoencoder architectures, as presented in Table 2 and Table 3. The decoders of both autoencoders were identical. Any neural network can extract features that differ from the features extracted using another neural network trained independently of the first, even if the training sets are identical. The correlation of features extracted by different neural networks is a desirable property.

It is known that the training of multilayer neural networks requires a large training set. However, the AIC-ears-75 database consists of only 2250 images, which is not enough for the training of autoencoders and subsequent testing of c-neuro-extractors.

Since the acoustic image of the ear has significant similarities with the voice signal, as well as their averaged spectra (Figure 11b–d), it was decided to use the following learning transfer scheme. From the speech datasets TIMIT (formed in 1993) and VoxCeleb1 [23], a total of 71,264 voice images of speakers were extracted. The sizes of the images ranged from 25 to 250 kbytes. Their averaged spectra were visually similar to the averaged spectra of the ear canal echograms using this size of sound files. However, the sampling frequency of the voice signals differed from the frequency characteristics of the ear canal echograms (16 kHz, the analyzed frequency range was up to 8 kHz, Figure 11b). These images were also converted to averaged spectra of 2048 amplitudes (with the parameters *W_size_* = 4096 and *W_step_* = 2048). In this case, only four window functions were used: Hamming, Blackman, triangular, and rectangular. Thus, 285,056 samples of averaged voice spectra, which were used to train two autoencoders, were obtained (Table 2 and Table 3). Neural networks were trained using the Adam optimization algorithm with an increasing batch size: 256 samples—10 epochs, 512 samples—10 epochs, 1024 samples—3 epochs, 2048 samples—3 epochs, 4096 samples—2 epochs, and 8192 samples—2 epochs. At the same time, intermediate quality control of the data recovery was provided (Figure 11e). The use of different window functions to obtain similar but different samples can be considered as augmentations of the training set data.

### 4.4. Autoencoder Training for Feature Extraction

A computational experiment was performed. The AIC-ears-75 dataset was randomly divided into two parts: 50 users (100 ears) were «Genuine Users» and 25 users (50 ears) were «Unseen Impostors». All images were processed by both encoders. In the first stage of the experiment, the images of the left and right ears were divided into two classes as if they were two different users (Table 4 and Table 5). In the second stage, the features extracted from both ears were combined into one (double) image (Table 6).

Samples of «Genuine» and «Impostors» were formed to train each c-neuro-extractor. The «Genuine» training set consisted of four to eight images (8 ≥ *K_G_* ≥ 4) of a specific user from the set of «Genuine Users», and the rest of the images of the user were used to test and calculate the FRR. The «Impostors» training set was formed from the images of other users from the «Genuine Users» set; one image of each ear was considered (*K_I_* = 99 in the first stage of the experiment and *K_I_* = 49 in the second, where *K_I_* is the number of images in the «Impostors» training set). Images from the set of «Unseen Impostors» were used only for testing and determining the FAR. The experimental results are presented in Table 4, Table 5 and Table 6.

A similar computational experiment was performed but used a base neuro-extractor model that was trained in accordance with GOST R 52,633.5. The best results are as follows:EER = 0.03041 (FRR = 0.2288 at FAR < 0.001) with a key length *L* = 716; the size of the «Genuine» training set was *K_G_* = 8 and the size of the «Impostors» training set was *K_I_* = 49.

## 5. Discussion

The feature vectors extracted from the averaged spectra of the same signal but based on different windows (Figure 11c,d) are strongly correlated (correlation coefficient ranged from 0.9 to 0.99). However, as we can see in Table 4, the use of strongly correlated features allows for a twofold reduction in the error probability and also a significant increase in the length of the associated key. We can also see that combining the features obtained using encoders that are slightly different in their architectures (Table 2 and Table 3) also has a positive effect (Table 5). For example, by combining the features extracted by both encoders based on the Hamming spectra, it is possible to reduce the error probability twofold (Table 4 and Table 5). In addition, when combining the features extracted by the encoders from the spectra based on the Hamming window and a rectangular window, the error probability decreases by more than 10%, whereas the key length increases by a factor of 2 (Table 4 and Table 5). The obtained results suggest that many similar feature extraction units (similar but slightly different nonlinear transformations with respect to the patterns/images) can be used to improve the performance of the c-neuro-extractor. This technique, as a rule, does not have such a tangible effect in combination with other machine learning methods (for most classifiers, these sets of features will be similar in information content). Of course, there is a limit to the reduction of errors, but this issue deserves a separate study.

A significant change in the number of neuron inputs is an unproductive approach (Table 5). The best results for the dataset used are achieved with η = 5 (Table 6). We can also conclude that it is not worth changing the boundaries of the intervals (*C_-_*; *C_+_*) significantly, and the optimal values are *C_-_* = −0.5, *C_+_* = 0.5 (Table 6). This can probably be explained by the fact that on the small «Genuine» training set (8 ≥ *K_G_* ≥ 4), the correlation coefficients are calculated with large errors. With an increase in *C_-_* and *C_+_* in the modulus, the set of pairs of correlated features that carry information does not fall into the specified intervals and is not used. When the modulus C_-_ and C_+_ are decreased, a lot of feature pairs that do not carry useful information are taken into account. There is also an optimum for AUC_MAX_≈0.3 (at least for the AIC-ears-75 dataset). Too large an AUC_MAX_ value negatively affects the results since more unstable neurons are created; too small a value leads to the creation of a small number of neurons.

Reducing the size of the «Genuine» training set does not greatly affect the probability of errors (Table 6). The model learns well with 8, 7, and 6 examples of images. Thus, the use of a large sample set for the c-neuro-extractor training is not required.

Figure 12 shows that if 16% of erroneous bits are corrected at the c-neuro-extractor output, then the authentication system can be configured for the following indicators: FAR < 0.001 at FRR = 0.093, which in general can be useful for practical purposes. The correction of the wrong key bits can be performed by classical methods of error-correcting coding (but unlike fuzzy extractors, this will not affect the key length). The obtained indicators are probably not limiting since the results can be improved by adding other window functions or several autoencoders with a different architecture (this is not the aim of the work). In any case, the proposed models are very promising for further development.

If necessary, the bit sequence arising at the output of the c-neuro-extractor can be transformed into a key of a desired length by applying the cryptographic hash function (after the correction of erroneous bits).

The comparison of the obtained results with those previously achieved is presented in Table 7. The c-neuro-extractor, as we can see, showed better results than those previously achieved.

## 6. Conclusions

The classical theory of mathematical statistics states that if features are correlated, they duplicate certain information. However, the obtained results indicate the opposite, that is, strongly correlated pairs of features contain additional information. Of course, a theory that has been proven for decades cannot be wrong. The data obtained only clarify it in relation to the problems of image (pattern) classification.

Correlation links between features deform the feature space. The nature of the curvature relative to each class of images is generally different since for different classes of images, the correlation matrices of features can differ significantly. This curvature makes it difficult to construct separating hyperplanes in the feature space in the process of machine learning. Independent (weakly correlated) informative meta-features can be extracted from correlated pairs of features. One meta-feature can contain 2–3 times more information than is contained in the pair of initial features from which it was generated (*I’_j,t_ > I_j_ + I_t_*). The new meta-feature space is called the Bayes–Minkowski meta-feature space.

A model of correlation neurons is proposed, which allows for the use of the Bayes–Minkowski meta-features for image (pattern) classification. The network of correlation neurons is trained automatically with a small training set and can be combined with a pre-trained deep neural network. This provides benefits both in terms of the simplification of the training procedure and information security. Correlation neuron networks are potentially more resistant to destructive influences such as adversarial attacks. When identifying an image that does not belong to any of the known classes, the networks of correlation neurons should generate an almost random binary code that should be ignored when making decisions. This work does not assert the superiority of networks of correlation neurons over multilayer networks, but only indicates the promising nature of correlation neurons in some respects (automatic learning/additional training, ensuring the safety of the decision-making process).

The proposed model of a c-neuro-extractor allows for the association of a cryptographic key or password with a user’s biometric image and the storage of both of these components safely without compromise. The proposed model surpasses the previously known models (fuzzy extractors, neuro-extractors) in key length while allowing a low percentage of obtained erroneous decisions. The experimental results showed high efficiency of the proposed model in the problems of key generation based on acoustic images of the ear: EER = 0.0238 (FRR = 0.093 at FAR < 0.001) with a key length *L* = 8192 bits; the volume of the «Genuine» training set was *K_G_* = 6 and that of the «Impostors» training set was *K_I_* = 49 (the following values were obtained for the base model of the neuro-extractor: EER = 0.03041, FRR = 0.2288, FAR < 0.001, *L* = 716, *K_G_* = 8, *K_I_* = 49).

There are many potential neuron constructs that allow for the analysis of correlations without compromising biometric templates. We can say that the Bayes–Minkowski measure is an “antagonist” in relation to the Minkowski measure since it has the opposite properties (makes fewer mistakes if the features are correlated and more errors if the features are independent). Therefore, they can be used together, by analyzing the strongly correlated features using the Bayes–Minkowski neurons and the weakly correlated ones with the Minkowski neurons. There are also plans to improve the learning algorithm for correlation networks.

In addition to biometric authentication, networks of correlation neurons can be used in other tasks of image (pattern) classification (also in combination with classical deep networks), especially in cases where the size of the training set is limited. The use of networks of correlation neurons for the synthesis of artificial intelligence resistant to adversarial attacks, as well as attacks aimed at extracting AI knowledge and various manipulations with AI models, seems promising.

## Figures and Tables

**Figure 1 sensors-22-09551-f001:**
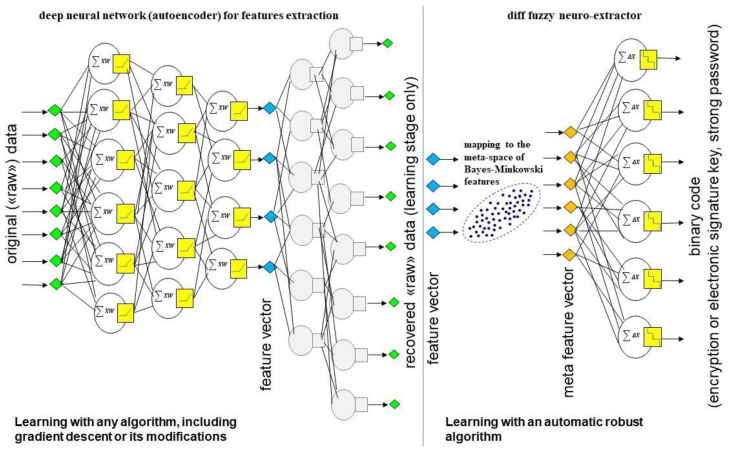
Block diagram of biometric-based key binding.

**Figure 2 sensors-22-09551-f002:**
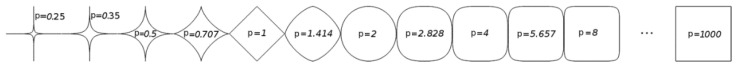
A circle on a plane at different values of the power coefficient *p*.

**Figure 3 sensors-22-09551-f003:**
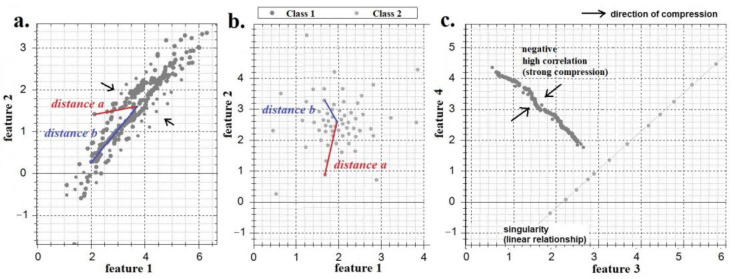
The direction of space compression of two features (**a**) with a positive significant correlation between features (the distance “a” is greater than the distance “b” since the feature space is not “flat” but curved due to the correlation); (**b**) with independent features (distance “a” is greater than “b”); (**c**) with different correlations.

**Figure 4 sensors-22-09551-f004:**
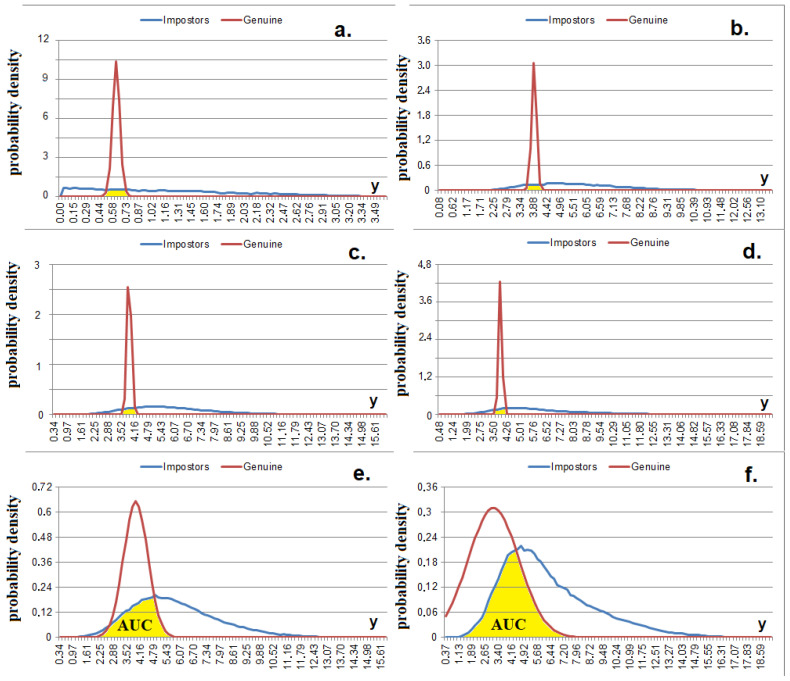
Probability densities of Measure (5) values (without square root) at *p* = 1, *I ≈ *1.75 bits. (**a**) for all classes where 1> *C_j,t_* > 0.95, *n’* = 1; (**b**) for all classes where 1 > *C_j,t_* > 0.95, *n’* = 5; (**c**) for the «Genuine» class where 1 > *C_j,t_* > 0.95, for the «Impostors» class where |*C_j,t_*| < 0.3, *n’* = 5; (**d**) for the «Genuine» class where 1 > *C_j,t_* > 0.95, for the «Impostors» class where −1 < *C_j,t_* < −0.95, *n’* = 5; (**e**) for all classes where |*C_j,t_*| < 0.3, *n’* = 5; (**f**) for all classes where −1 < *C_j,t_* < −0.95, *n’* = 5.

**Figure 5 sensors-22-09551-f005:**
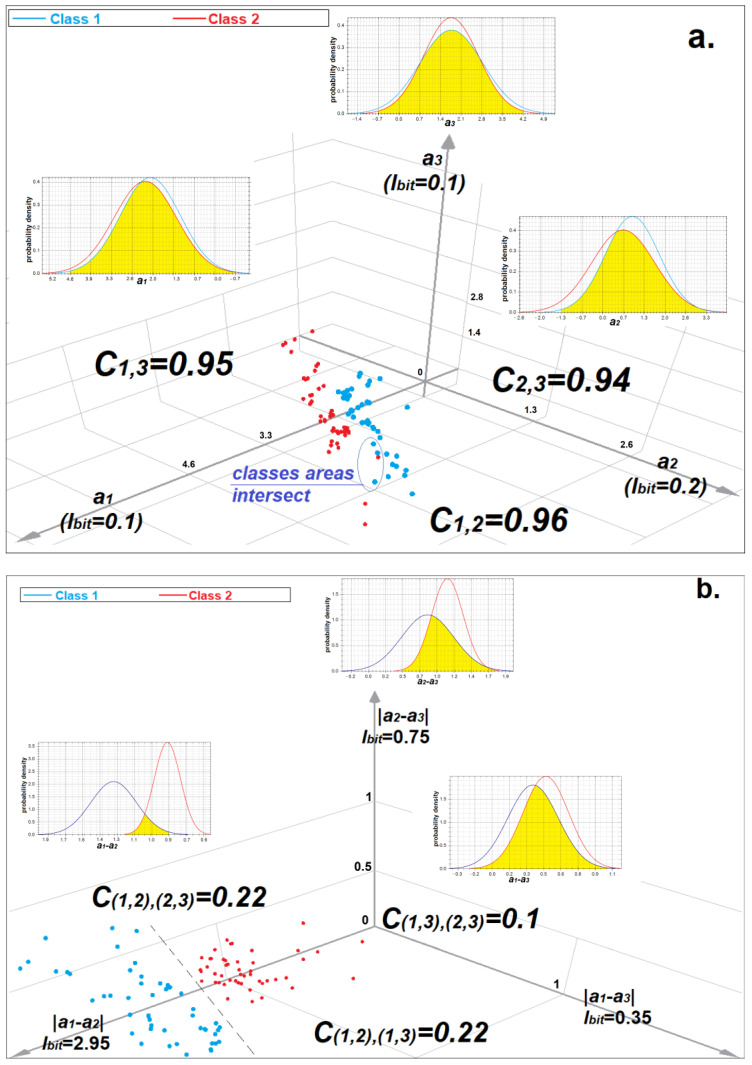
Spaces of three features (meta-features) and plots of the probability densities of their values: (**a**) the initial space of positively correlated features; (**b**) space of Bayes–Minkowski meta-features derived from the initial feature space by applying mapping (12) with *p* = 1.

**Figure 6 sensors-22-09551-f006:**
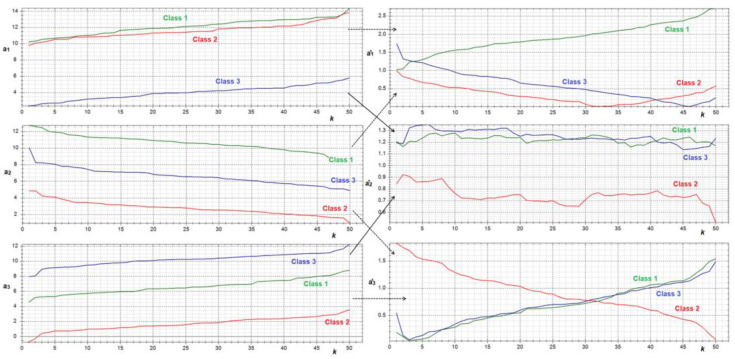
Initial features (left) and meta-features (right) generated by mapping (12) *p* = 0.

**Figure 7 sensors-22-09551-f007:**
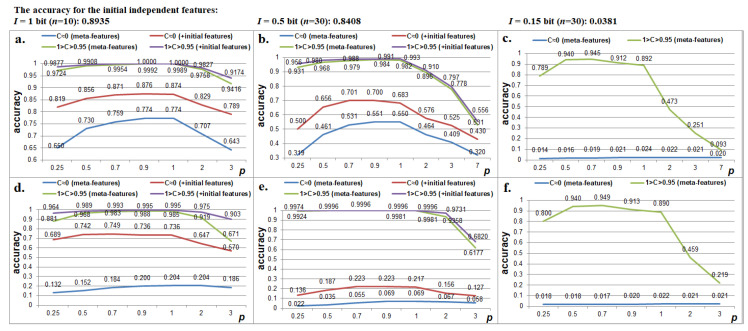
Changing the accuracy of the identification of images based on the “naive” Bayes classifier with various average informativeness of the initial features, when applying the mapping in (12): (**a**) *I* ≈ 1 bits, *n* = 10 and *n’* = 45; (**b**) *I* ≈ 0.5 bits, *n* = 30 and *n’* = 435; (**c**) *I* ≈ 0.15 bits, *n* = 30 and *n’* = 435, and when applying the mapping in (11): (**d**) *I* ≈ 1 bits, *n* = 10 and *n’* = 45; (**e**) *I* ≈ 0.5 bits, *n* = 30 and *n’* = 435; (**f**) *I* ≈ 0.15 bits, *n* = 30 and *n’* = 435.

**Figure 8 sensors-22-09551-f008:**
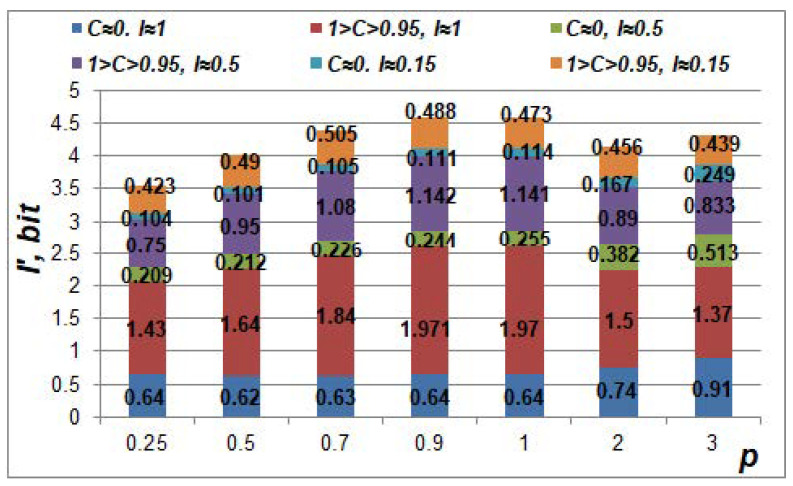
Average informativeness of meta-features *I’* generated by the mapping in (12).

**Figure 9 sensors-22-09551-f009:**
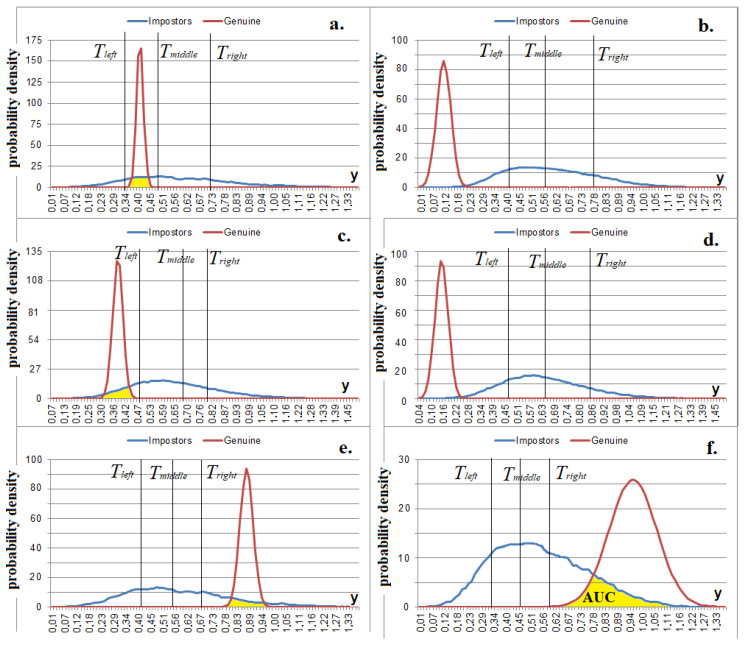
Plots of probability densities of measured values in (16) after the mapping in (12) at *p* = 1, *I* ≈ 1.75 bits (**a**) for all classes where 1 > *C_j,t_* > 0.95, *n’* = 10; (**b**) for all classes where −1 < *C_j,t_* < −0.95, *n’* = 10; (**c**) for the «Genuine» class where 1 > *C_j,t_* > 0.95, for the «Impostors» class where |*C_j,t_*| < 0.3, *n’* = 10; (**d**) for the «Genuine» class where −1 < *C_j,t_* < −0.95, for the «Impostors» class where |*C_j,t_*| < 0.3, *n’* = 10; (**e**) for the classes where 1 > *C_j,t_* > 0.95, *n’* = 10 («Genuine» class is located on the right); (**f**) for the classes where −1 < *C_j,t_* < −0.95, *n’* = 10 («Genuine» class is located on the right).

**Figure 10 sensors-22-09551-f010:**
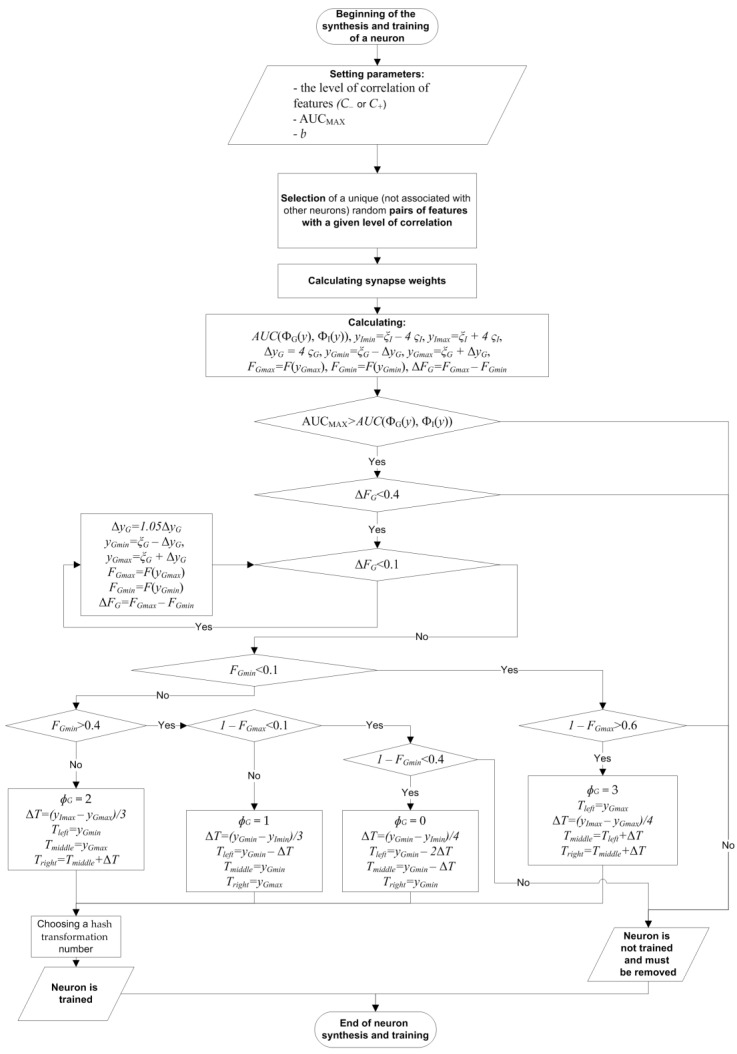
Scheme of the algorithm of synthesis and training of the correlation neuron.

**Figure 11 sensors-22-09551-f011:**
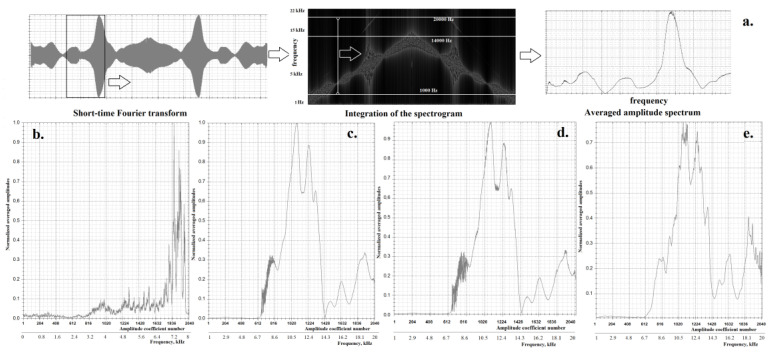
Average spectrum: (**a**) calculation process; (**b**) extracted from a voice image (from the TIMIT database) based on a rectangular window; (**c**) extracted from the acoustic image of the ear at the base of the Hamming window; (**d**) extracted from the acoustic image of the ear based on a rectangular window; (**e**) extracted from an acoustic image of an ear based on a Hamming window and reconstructed by an autoencoder.

**Figure 12 sensors-22-09551-f012:**
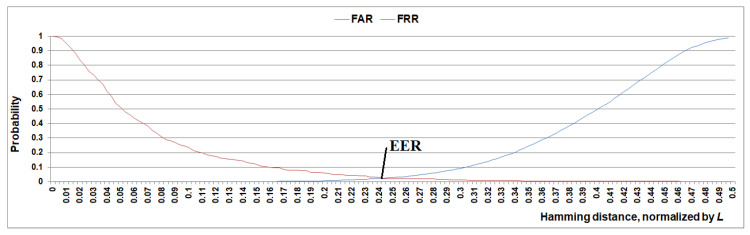
Results of user verification using two ears at C_+_ = 0.5, C_-_ = 0.5, AUC_MAX_ = 0.3, K_G_ = 6, N = 4096, η = 5.

**Table 1 sensors-22-09551-t001:** Variants of the hashing transformations of the neuron response *ɸ(y)* into a binary code.

Hash Transformation №	*ɸ(y)*			
	*0*	*1*	*2*	*3*
**1**	«11»	«00»	«01»	«10»
**2**	«11»	«00»	«10»	«01»
**3**	«11»	«01»	«00»	«10»
**4**	«11»	«01»	«10»	«00»
**5**	«11»	«10»	«00»	«01»
**6**	«11»	«10»	«01»	«00»
**7**	«00»	«01»	«11»	«10»
**8**	«00»	«01»	«10»	«11»
**9**	«00»	«10»	«01»	«11»
**10**	«00»	«10»	«11»	«01»
**11**	«00»	«11»	«10»	«01»
**12**	«00»	«11»	«01»	«10»
**13**	«01»	«00»	«11»	«10»
**14**	«01»	«00»	«10»	«11»
**15**	«01»	«10»	«00»	«11»
**16**	«01»	«10»	«11»	«00»
**17**	«01»	«11»	«10»	«00»
**18**	«01»	«11»	«00»	«10»
**19**	«10»	«00»	«01»	«11»
**20**	«10»	«00»	«11»	«01»
**21**	«10»	«01»	«11»	«00»
**22**	«10»	«01»	«00»	«11»
**23**	«10»	«11»	«00»	«01»
**24**	«10»	«11»	«01»	«00»

**Table 2 sensors-22-09551-t002:** Configurations of encoders.

Architecture №1		Architecture №2	
Layer Type	Layer Parameters	Layer Type	Layer Parameters
Input	shape = 2048	Input	shape = 2048
Conv1D	filters = 8, kernel_size = 12, strides = 4, activation = relu, initializer = glorot	Conv1D	filters = 8, kernel_size = 9, strides = 2, activation = relu, initializer = he
Conv1D	filters = 16, kernel_size = 3, strides = 2, activation = relu, initializer = glorot	Conv1D	filters = 16, kernel_size = 3, strides = 2, activation = relu, initializer = he
Batch normalization		Batch normalization	
Conv1D	filters = 16, kernel_size = 4, strides = 2, activation = relu, initializer = glorot	Conv1D	filters = 16, kernel_size = 4, strides = 2, activation = relu, initializer = he
Conv1D	filters = 32, kernel_size = 3, strides = 2, activation = relu, initializer = glorot	Conv1D	filters = 32, kernel_size = 3, strides = 2, activation = relu, initializer = he
Batch normalization		Batch normalization	
Conv1D	filters = 32, kernel_size = 3, strides = 2, activation = relu, initializer = glorot	Conv1D	filters = 32, kernel_size = 3, strides = 1,2, activation = relu, initializer = he
Conv1D	filters = 64, kernel_size = 3, strides = 2, activation = relu, initializer = glorot	Conv1D	filters = 64, kernel_size = 3, strides = 2, activation = relu, initializer = he
Batch normalization		Batch normalization	
Conv1D	filters = 64, kernel_size = 3, strides = 2, activation = relu, initializer = glorot	Conv1D	filters = 64, kernel_size = 3, strides = 2, activation = relu, initializer = he
Conv1D	filters = 64, kernel_size = 3, strides = 2, activation = relu, initializer = glorot	Conv1D	filters = 64, kernel_size = 3, strides = 2, activation = relu, initializer = he
Batch normalization		Batch normalization	
Conv1D	filters = 128, kernel_size = 3, strides = 2, activation = relu, initializer = glorot	Conv1D	filters = 128, kernel_size = 3, strides = 2, activation = relu, initializer = he
Conv1D	filters = 256, kernel_size = 3, strides = 2, activation = relu, initializer = glorot	Conv1D	filters = 160, kernel_size = 3, strides = 2, activation = relu, initializer = he
Fully connected	units = 128, activation = linear, initializer = glorot	Batch normalization	
		Conv1D	filters = 192, kernel_size = 3, strides = 2, activation = relu, initializer = he
		Fully connected	units = 128, activation = linear, initializer = glorot

**Table 3 sensors-22-09551-t003:** Configurations of decoders.

Layer Type	Layer Parameters
Input	shape = 128
Conv1D Transpose	filters = 160, kernel_size = 8, strides = 4, activation = relu, initializer = glorot
Conv1D Transpose	filters = 128, kernel_size = 3, strides = 2, activation = relu, initializer = glorot
Batch normalization	
Conv1D Transpose	filters = 64, kernel_size = 5, strides = 2, activation = relu, initializer = glorot
Conv1D Transpose	filters = 32, kernel_size = 3, strides = 2, activation = relu, initializer = glorot
Batch normalization	
Conv1D Transpose	filters = 32, kernel_size = 3, strides = 2, activation = relu, initializer = glorot
Conv1D Transpose	filters = 16, kernel_size = 3, strides = 2, activation = relu, initializer = glorot
Batch normalization	
Conv1D Transpose	filters = 8, kernel_size = 3, strides = 2, activation = relu, initializer = glorot
Conv1D Transpose	filters = 8, kernel_size = 3, strides = 2, activation = relu, initializer = glorot
Batch normalization	
Conv1D Transpose	filters = 4, kernel_size = 3, strides = 2, activation = relu, initializer = glorot
Conv1D Transpose	filters = 1, kernel_size = 3, strides = 2, activation = sigmoid, initializer = glorot

**Table 4 sensors-22-09551-t004:** Error probabilities of user personality verification with the image of one ear using features extracted by the encoder based on architectures 1 and 2 (with *C_+_* = 0.5, *C_-_* = 0.5, AUC_MAX_ = 0.3, *K_G_* = 8).

Type of Spectra	*L*	*N*	*n*	*η*	EER_1_	EER_2_
Blackman	128	64	128	4	0.0863	0.08828
Hamming	128	64	128	4	0.08747	0.07332
Triangular	128	64	128	4	0.09133	0.08995
Rectangular	128	64	128	4	0.08122	0.07686
Gauss	128	64	128	4	0.09047	0.08039
Gaussian parametric	128	64	128	4	0.08465	0.07629
Laplace	128	64	128	4	0.07452	0.08589
Rectangular + Hamming	128	64	256	4	0.04351	0.04272
Rectangular + Hamming + triangular	256	128	384	4	0.04245	0.04218
Rectangular + Hamming + triangular + Laplace + Blackman	512	256	640	4	0.04061	0.03812
All window functions	512	256	896	4	0.04031	0.03574

**Table 5 sensors-22-09551-t005:** Error probabilities of user personality verification with the image of one ear using combined features extracted by both encoders (at *C_+_* = 0.5, *C_-_* = 0.5, AUC_MAX_ = 0.3, *K_G_* = 8).

Type of Spectra	*L*	*N*	*n*	*η*	EER
Hamming	128	64	256	4	0.03591
rectangular + Hamming	256	128	512	5	0.03823
rectangular + Hamming + triangular + Laplace + Blackman	512	256	1280	5	0.03474
All window functions	512	256	1792	5	0.03955
All window functions	1024	512	1792	5	0.0366
All window functions	1024	512	1792	20	0.03834
All window functions	2048	1024	1792	5	0.03466
All window functions	2048	1024	1792	10	0.03635
All window functions	4096	2048	1792	5	0.03573
All window functions	4096	2048	1792	10	0.04124

**Table 6 sensors-22-09551-t006:** Error probabilities of user personality verification with the images of two ears using the combined features of all types of spectra extracted by both encoders.

*L*	*N*	*n*	*η*	EER	*C_+_*	*C_-_*	*K_G_*	AUC_MAX_
4096	2048	3584	5	0.02865	0.5	−0.5	8	0.3
6144	3072	3584	5	0.02811	0.5	−0.5	8	0.3
8192	4096	3584	7	0.02956	0.5	−0.5	8	0.3
8192	4096	3584	6	0.02584	0.5	−0.5	8	0.3
*8192*	*4096*	*3584*	*5*	*0.02552*	*0.5*	*−0.5*	*8*	*0.3*
8192	4096	3584	4	0.02561	0.5	−0.5	8	0.3
8192	4096	3584	3	0.0273	0.5	−0.5	8	0.3
10,240	5120	3584	5	0.02729	0.5	−0.5	8	0.3
12,288	6144	3584	5	0.02725	0.5	−0.5	8	0.3
8192	4096	3584	5	0.03025	0.5	−0.5	8	0.4
8192	4096	3584	5	0.02878	0.5	−0.5	8	0.35
8192	4096	3584	5	0.02703	0.5	−0.5	8	0.25
8192	4096	3584	5	0.03673	0.5	−0.5	8	0.2
8192	4096	3584	5	0.02619	0.3	−0.3	8	0.3
8192	4096	3584	5	0.026	0.4	−0.4	8	0.3
8192	4096	3584	5	0.02563	0.45	−0.45	8	0.3
8192	4096	3584	5	0.02754	0.55	−0.55	8	0.3
8192	4096	3584	5	0.02855	0.6	−0.6	8	0.3
8192	4096	3584	5	0.03274	0.7	−0.7	8	0.3
8192	4096	3584	5	0.02785	0.5	−0.5	7	0.3
* **8192** *	* **4096** *	* **3584** *	* **5** *	* **0.0238** *	* **0.5** *	* **−0.5** *	* **6** *	* **0.3** *
8192	4096	3584	5	0.03868	0.5	−0.5	5	0.3
8192	4096	3584	5	0.04098	0.5	−0.5	4	0.3

**Table 7 sensors-22-09551-t007:** Results of personality recognition by the ear echographic.

Feature Extraction Method	Classification Method	Data Set	Error Probabilities	
Filter of saddle-point method	Determination of the correlation coefficient between the template and the image	Mobile phone: 17 test subjects, 8 images each	0.055 ≤ EER ≤ 0.18 [24]	
		In-ear headphones: 31 test subjects, 8 images each	0.01 ≤ EER ≤ 0.06 [24]	
		On-ear headphones: 31 test subjects, 8 images each	0.008 ≤ EER ≤ 0.08 [24]	
Short-time Fourier transform	k-nearest neighbors	20 test subjects, 600 images each, 11,900 images in total	0.05 ≤ FRR ≤ 0.1 0.07 ≤ FAR ≤ 0.15 [25]	
	Decision trees		0.09 ≤ FRR ≤ 0.152 0.09 ≤ FAR ≤ 0.145 [25]	
	Naive Bayes		0.03 ≤ FRR ≤ 0.08 0.09 ≤ FAR ≤ 0.275 [25]	
	Multilayer perceptron		0.03 ≤ FRR ≤ 0.098 0.04 ≤ FAR ≤ 0.08 [25]	
	Support vector machine		0.04 ≤ FRR ≤ 0.07 0.03 ≤ FAR ≤ 0.07 [25]	
Author’s technology EarEcho			EER = 0.0484 [25]	
MFCCs, LDA	Cosine similarity	25 test subjects	EER = 0.0447 [26]	
Cepstrograms	Naive Bayes	AIC-ears-75 75 test subjects, 15 images for each ear, 2250 images in total	«Genuine» training set of 8 images	EER = 0.0053 FRR = 0.1028 FAR < 0.001 [1]
	Convolutional neural networks			EER = 0.0285 [1]
Average spectrum	Fully connected shallow neural networks			EER = 0.0266 [1]
Average spectrum, training of autoencoders on voice images	Base model of neuro-extractor (GOST R 52,633.5)		«Genuine» training set of 8 images	EER = 0.03041 FRR = 0.2288 FAR < 0.001
	C-neuro-extractor		«Genuine» training set of 6 images	EER = 0.0238 FRR = 0.093 FAR < 0.001

## Data Availability

The dataset of the ear acoustic images (AIC-ears-75) is available at http://aiconstructor.ru/page14247028.html (accessed on 25 November 2022). You can also contact the author to obtain a sample of the trained autoencoders and data that were used to train autoencoders.

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
