# Peer review of "Biometric-Based Key Generation and User Authentication Using Acoustic Characteristics of the Outer Ear and a Network of Correlation Neurons"

_sensors, 2022, doi:10.3390/s22239551_

Round 1

Reviewer 1 Report

The contribution of the paper is high; however, I am afraid the paper might not be readable for a general reader. Authors may introduce some explanation of the terms before use which is highly specific to this domain.

Anyway, good work.

Author Response

I express my sincere gratitude to the respected reviewer! Thanks for high appreciating the work and for the valuable remark, which was fixed.

Section 2. Related Works was expanded and named 2. Related Works. Common Terms. Additional terminology clarifications have been added to it. I hope this will help make the article more understandable.

Thanks again!

Reviewer 2 Report

-The abstract should be rewritten to give and idea of the context before of the method proposed. At the moment it seems to be a list of claims about the paper and not a brief summary of the work.

-The language needs to be improved .

-The style of the references should be unified.

-Formulas should be corrected. As an example, in (1) "a signed" is defined but not present in the formula. 

-Figure 3 contains too much details, it should be simplified. 

-Algorithm in Figure 10 should be better presented as pseudo-code.

-Please adjust Table 1. In its present form it is not legible.

Author Response

I express my gratitude to the respected reviewer for reviewing the article and for the remarks that really helped to make the article better!

Briefly about the introduced changes:

-The abstract should be rewritten to give and idea of the context before of the method proposed. At the moment it seems to be a list of claims about the paper and not a brief summary of the work.

Fixed. Abstract rewritten in accordance with the recommendations on the journal's website

-The language needs to be improved .

Fixed. The article was reviewed and corrected by an experienced translator, I hope we succeeded

-The style of the references should be unified.

Fixed.

-Formulas should be corrected. As an example, in (1) "a signed" is defined but not present in the formula. 

Fixed. I'm not sure that I fully understood this remark, but it has also been corrected in accordance with our understanding. We hope that the corrections made will suit you.

-Figure 3 contains too much details, it should be simplified. 

Fixed.

-Algorithm in Figure 10 should be better presented as pseudo-code.

Fixed. Pseudo-code added to Appendix A. Figure 10 we also decided to keep

-Please adjust Table 1. In its present form it is not legible.

Fixed.

Once again, thank you very much!

Reviewer 3 Report

The article is well written and gives a detailed presentation of the proposed solution.

A minor observation: the author should provide further explanation related to the similarity between voice signals and acoustic images. The average spectra shown in figure 11 are partially relevant because the voice spectrum stops at 8KHz thus covering only a small part of the acoustic image spectrum which has most of the energy located in the 8-20KHz range.

In conclusion, I recommend accepting the article with minor revision.

Author Response

I express my sincere gratitude to the respected reviewer! Thanks for high appreciating the work and for the valuable remark, which was fixed.

The fact is that the sampling frequency of voice signals differed from the "ear" ones (16 kHz versus 44 kHz). Therefore, voice signals were processed with a different Fourier window length. However, after processing, both types of signals were reduced to an identical number of average spectrum coefficients (2048). And these spectra really had some visual similarity, despite the different frequency characteristics of the original signals. For the autoencoder, there is no difference, since the input is vectors of 2048 values ​​in both cases.

We have made adjustments to figure 11 (adding a double scale on the x-axis) and some additional clarifications in the text.

Thanks again!

Reviewer 4 Report

The current manuscript presents a method of adopting ear canal as a biometric based key, with the aid of the acoustic imaging. The author has shown his in-depth knowledge of the machine learning with his methodology explained in detail. All parameters for the acoustic imaging experiments are shown. This should be appreciated. The manuscript is overall well-organized. My only concern is about figure 5 that its presentation is confusing with other figures rotated and embedded in

Author Response

I express my sincere gratitude to the respected reviewer! Thanks for high appreciating the work and for the valuable remark, which was fixed.

We have tried to convey Figure 5 as far as possible.

Once again, thank you very much!